# Opposite Nuclear Dynamics of Two FRH-Dominated Frequency Proteins Orchestrate Non-Rhythmic Conidiation in *Beauveria bassiana*

**DOI:** 10.3390/cells9030626

**Published:** 2020-03-05

**Authors:** Sen-Miao Tong, Ding-Yi Wang, Qing Cai, Sheng-Hua Ying, Ming-Guang Feng

**Affiliations:** 1College of Agricultural and Food Science, Zhejiang A&F University, Lin’an 311300, Zhejiang, China; 2MOE Laboratory of Biosystems Homeostasis & Protection, Institute of Microbiology, College of Life Sciences, Zhejiang University, Hangzhou 310058, Zhejiang, China

**Keywords:** filamentous fungi, entomopathogenic fungi, mutagenesis, gene expression and regulation, asexual development, conidiation, photoperiod response

## Abstract

Non-rhythmic conidiation favors large-scale production of conidia serving as active ingredients of fungal insecticides, but its regulatory mechanism is unknown. Here, we report that two FREQUENCY (FRQ) proteins (Frq1/2) governed by a unique FRQ-interacting RNA helicase (FRH) orchestrate this valuable trait in *Beauveria bassiana*, an asexual insect-pathogenic fungus. Frq1 (964 aa) and Frq2 (583 aa) exhibited opposite expression dynamics (rhythms) in nucleus and steadily high expression levels in cytoplasm under light or in darkness no matter whether one of them was present or absent. Such opposite nuclear dynamics presented a total FRQ (pooled Frq1/2) level sufficient to persistently activate central developmental pathway in daytime and nighttime and supports continuous (non-rhythmic) conidiation for rapid maximization of conidial production in a fashion independent of photoperiod change. Importantly, both nuclear dynamics and cytoplasmic stability of Frq1 and Frq2 were abolished in the absence of the FRH-coding gene nonessential for the fungal viability, highlighting an indispensability of FRH for the behaviors of Frq1 and Frq2 in both nucleus and cytoplasm. These findings uncover a novel circadian system more complicated than the well-known *Neurospora* model that controls rhythmic conidiation, and provide a novel insight into molecular control of non-rhythmic conidiation in *B. bassiana*.

## 1. Introduction

Aerial conidiation is critical for survival and dispersal in host habitats of filamentous fungal pathogens. This process is genetically controlled by the upstream development activation (UDA) pathway comprising signal transducers (FluG and FlbA–E) and the central developmental pathway (CDP) consisting of three sequentially activated genes (*brlA*, *abaA*, and *wetA*) in some filamentous fungi, such as Eurotiales [1] (also reviewed in [2,3]). *Beauveria bassiana* is a filamentous fungal insect pathogen that undergoes asexual cycle with teleomorph never induced successfully under laboratory conditions and serves as a main source of fungal insecticides and acaricides containing formulated conidia as active ingredients (reviewed in [4,5]). This pathogen features the broadest host range among all fungal pathogens known to date despite an intraspecific variation in conidiation capacity and virulence (reviewed in [6]). In *B. bassiana*, conidiation and conidial maturation are regulated by the CDP genes as seen in model fungi, such as *Aspergillus* and *Penicillium* [2,3], because conidial production was abolished in the absence of *brlA* or *abaA* [7] and nearly abolished in the absence of *wetA* [8]. However, it remains unclear how CDP is activated in *B. bassiana* and also why the insect pathogen can produce conidia rapidly in a non-rhythmic fashion and maximize conidial production within a short period under optimal conditions. For example, non-rhythmic conidiation is faster in longer daylight lengths and leads to a maximal yield of ~5 × 10^8^ conidia cm^−2^ culture surface within 7 to 9 days of incubation at the regimes of optimal 25 °C with a photoperiod change from full dark to light [9]. In *B. bassiana*, severe or extremely severe conidiation defects caused by knockout mutations of many genes often correlate with drastic repression of *brlA*, *abaA*, and *wetA*, such as those encoding the Na^+^/H^+^ antiporter Nhx1 [10], the vacuolar protein VLP4 [11], the blue light sensor VIVID (VVD) [9], the histone acetyltransferases Gcn5 and Mst2 [12,13], the histone deacetylases Hos2 and Rpd3 [14,15], and the mitogen-activated protein kinase (MAPK) kinase Ste7 [16]. Particularly, Gcn5 has been shown to activate the *brlA*-hallmarked CDP by catalyzing histone H3K14 phosphorylation at the *brlA* promoter region [12]. These studies imply possible existence of multiple routes responsible for transcriptional activation of the CDP genes in *B. bassiana*.

Contrasting to non-rhythmic conidiation in *B. bassiana*, rhythmic production of orange (carotenoid-pigmented) conidia in *Neurospora crassa* can be readily traced in a self-reporting race assay and is controlled precisely by a transcription-translation feedback loop that consists of an activating positive arm and a repressing negative arm and acts as an circadian clock [17], as reviewed in [18,19]. In the positive arm, two White Collar proteins (WC-1 and WC-2) form heterodimeric White Collar complex (WCC), a transcription factor that activates expression of the negative-arm protein FREQUENCY (FRQ) [20,21,22]. Upon activation, FRQ interacts with FRQ-interacting RNA helicase (FRH) and casein kinase 1 (CK1), forming the FRQ-FRH complex (FFC) that interacts with the WCC [23]. These interactions critical for timekeeping of the feedback loop occur in the nucleus where FRQ functions as a key oscillator of the clock [24]. In *N. crassa*, a single *frq* gene encodes long and short FRQ isoforms (l-FRQ and s-FRQ) with a difference of only 99 amino acids (aa) in molecular size [25], and the ratio of the two isoforms is regulated by thermosensitive splicing at different codons [26,27,28]. Transcription of *frq* in the nucleus is induced by the WCC binding to the light-responsive and circadian DNA elements in the *frq* promoter [20,21,29,30], followed by FRQ synthesis for formation of functional homodimers through a coiled-coil domain [31,32]. Consequently, the homodimers enter the nucleus to rapidly repress *frq* transcription [24,33,34,35,36] via FRQ/FRH-WCC interaction and then FRQ/FRH-promoted phosphorylations of the WCC that remove the latter from the DNA [35,37,38,39,40]. However, it remains opaque whether FRH is an essential or nonessential player in the negative arm due to its essentiality for cell viability of *N. crassa*. FRH was first revealed to be essential for the negative arm by analysis of knockdown *frh* mutants, which showed reduced FRQ level, increased *frq* RNA level and abolished circadian rhythm [23]. A small *C*-terminal region of FRQ was found to act as an essential domain for formation of the FFC required for both the FRQ-WCC interaction and the clock function [41]. The domain was also shown to play a major role in nuclear localization of FRQ [42]. On the other hand, the site-mutated allele *frh^R860H^* to bypass the essentiality of *frh* for the fungal viability was found to impair the clock function of *N. crassa* due to its effect on the WCC/FRQ-FRH^R860H^ interaction but no impact on the FRQ-FRH^R860H^ interaction [43]. Further mutational analysis uncovered nonessential role of FRH in the fungal clock despite its essentiality for stabilization of intrinsically disordered FRQ by preventing FRQ from rapid degradation [44,45]. The nonessential role of FRH in the clock is further revealed by structural analysis of FRH crystals, including R860H and V142G substitutions that disturbed the clock function or altered the activities of FRQ and WCC binding to FRH [46]. The previous studies on the mutants of *frh* knockdown or site-mutated alleles unveiled interactions of FRH, with both FRQ and WCC essential for the clock function. However, the performances of such mutants bypassing a requirement of *frh* for cell viability may not reflect complete role of FRH in the feedback loop. Since nuclear localization is required for FRQ to act as the clock oscillator [24], we speculate that daily nuclear dynamics of FRQ may directly reflect its own circadian rhythm linked to certain traceable or untraceable phenotypes and can be exploited to clarify essential or nonessential role of FRH in the feedback loop if the whole FRH function is controllable.

Filamentous fungi are manifold (saprophytic to pathogenic) in lifestyle, and most of them lack a traceable phenotype to self-report circadian rhythm, as seen in *N. crassa*. As far as learned to date, the key oscillator FRQ is absent in *Aspergillus* but a clock exists in *A. flavus* to mediate a 33-h rhythm in the development of sclerotia critical for environmental survival and in *A. nidulans* to regulate a free running rhythm in the mRNA accumulation of glyceraldehyde-3-phosphate dehydrogenase (GpdA) gene [47]. *Botrytis cinerea* has clock components similar to those in *N. crassa*, including FRQ (BcFRQ1), to not only regulate circadian rhythm of the fungal infection but also function in a non-circadian fashion [48]. In *Magnaporthe oryzae*, *frq* transcription cannot be activated by *wc-1* knockout mutation, which compromised light-dependent disease suppression [49]. A survey of 64 fungal proteomes has revealed existence of up to 10 FRQ-like proteins in *Fusarium oxysporum* strains [50]. These studies suggest diversity and complexity of circadian systems functioning in different lineages of filamentous fungi aside from the *Neurospora* model.

In a survey of the *B. bassiana* genome [51], we found two distinct FRQ homologs (named Frq1 and Frq2), a unique FRH, and other putative clock proteins, including two WCs, VVD, and the red/far-red light sensor phytochrome (PHY). Previously, VVD was shown to migrate from cytoplasm to nucleus in response to short daylight length and to regulate conidiation level in a daylight length-dependent manner [9]. PHY was also found to mediate conidial yield in a similar fashion [52]. We hypothesize that Frq1 and Frq2 in *B. bassiana* may have their own rhythms in nucleus to support non-rhythmic conidiation and continuous asexual cycle in a fashion independent of seasonal or geographic photoperiod change. This study seeks to test the hypothesis by monitoring subcellular localization of Frq1 and Frq2 and their nuclear and cytoplasmic expression dynamics in time-course responses to light and dark cues. Our goal is to uncover whether non-rhythmic conidiation is orchestrated by Frq1 and Frq2 and whether FRH is essential or non-essential for the roles of both FRQs in *B. bassiana*.

## 2. Materials and Methods

### 2.1. Identification and Bioinformatic Analysis of FRQ and FRH Proteins in B. bassiana

The sequences of l-FRQ (NCBI accession code: AAA57121), s-FRQ (ESA42015), and FRH (EAA27062) in *N. crassa* were used as queries to search through the *B. bassiana* genome [51] at http://blast.st-va.ncbi.nlm.nih.gov/Blast.cgi. The located FRQ homologs, namely Frq1 (EJP69563) and Frq2 (EJP62033), were structurally compared with the queries through domain analysis at https://www.ncbi.nlm.nih.gov/Structure or http://smart.embl-heidelberg.de. Nuclear localization signal (NLS) was predicted from each FRQ at http://nls-mapper.iab.keio.ac.jp/. The two FRQs and their homologs found in other filamentous fungi were subjected to phylogenetic analysis with a neighbor-joining method in MEGA7 at http://www.megasoftware.net. Both the domain analysis and sequence alignment analysis with DNAman 8.0 (http://www.bio-soft.net/format/DNAMAN.htm) were performed to compare an amino acid sequence of the located FRH ortholog (EJP68494) with that of the query.

### 2.2. Generation of *frq1*, *frq2*, and *frh* Mutants

The genes *frq1* (tag locus: BBA_01528), *frq2* (BBA_08957), and *frh* (BBA_02496) were deleted from the wild-type strain *B. bassiana* ARSEF 2860 (designated WT herein) and rescued in an identified deletion mutant of each gene using a markerless protocol, which is based on a uridine auxotrophy (Δ*ura3*) of WT [53]. Briefly, the 5′ and 3′ coding/flanking fragments of *frq1* (1360 and 1406 bp), *frq2* (1416 and 1488 bp) or *frh* (1490 and 1384 bp) were cloned from the genomic DNA of WT with two pairs of primers and inserted into appropriate restriction sites in p0380-ura3 [54,55], forming p0380-5′*x*-ura3-3′*x* (*x* = *frq1*, *frq2* or *frh*) for targeted gene deletion. The full-length coding/flanking sequence of *frq1* (7099 bp), *frq2* (5613 bp), or *frh* (4789 bp) was amplified from the WT DNA and ligated into p0380-sur-gateway [56] to exchange for the gateway fragment, yielding p0380-sur-*x* for targeted gene complementation. Each deletion plasmid was transformed into Δ*ura3* for homogeneous recombination to delete each target gene and complement *ura3* via *Agrobacterium*-mediated transformation. The deleted gene was complemented by ectopic integration of p0380-sur-*x* into an identified deletion mutant with the same transformation method. Putative mutant colonies were screened based on an ability for the deletion mutants to grow in the absence of exogenous uridine and the *sur* resistance of the rescued mutants to chlorimuron ethyl (10 μg mL^−1^). Expected recombination events were verified via PCR and southern blot analyses. All of the paired primers used for targeted gene manipulation and mutant identification are listed in Appendix A. Positive deletion and complementation mutants of each target gene (Appendix A) were used in the study wherever necessary.

#### Subcellular Localization of Frq1, Frq2, and FRH

The coding sequences of *frq1* (2968 bp), *frq2* (1753 bp), and *frh* (3297 bp) were amplified from the WT cDNA with paired primers (Appendix A) and fused to the *N*-terminus of *gfp* (GenBank ID: U55763) encoding green fluorescence protein (GFP), respectively. The fusion genes were digested with appropriate enzymes and individually inserted into linearized pAN52-bar [57]. The new plasmids pAN52-*x*-bar (*x* = *frq1*::*gfp*, *frq2*::*gfp*, or *frh*::*gfp*) were linearized with NotI and integrated into the WT strain via blastospore transformation [57]. Positive transgenic colonies were screened by the *bar* resistance to phosphinothricin (200 μg mL^−1^). Transgenic strains best expressing GFP-tagged Frq1, Frq2, and FRH fusion proteins, namely TS1, TS2, and TS3 respectively, were chosen for subcellular localization. The selected strains were incubated at the optimal regime of 25 °C in a light/dark (L:D) cycle of 12:12 h on SDAY [Sabouraud dextrose agar (4% glucose, 1% peptone and 1.5% agar) plus 1% yeast extract] until full conidiation. The resultant conidia were cultivated in SDBY (i.e., agar-free SDAY) by shaking (150 rpm) 3 days at 25 °C in the L:D cycles of 0:24, 12:12, and 24:0. Hyphae were collected from the cultures, stained with the membrane-specific dye FM4-64 or the nuclear dye DAPI (Sigma, St. Louis, MO, USA), and visualized with laser scanning confocal microscopy (LSCM). Subcellular localization of each fusion protein in each L:D cycle was judged based on the LSCM images.

To verify whether each fusion protein was expressed in a correct form and in nucleus and/or cytoplasm, total, cytoplasmic and nuclear protein extracts were isolated from the 3-day-old SDBY cultures of the transgenic strains grown respectively in the L:D cycles of 24:0, 12:12, and 0:24 as described previously [9,58] or following the manufacturer’s guide of Minute™ Cytoplasmic and Nuclear Extraction Kits (Invent Biotechnologies, Inc., Plymouth, MN, USA; Catalog No.: SC-003). Particularly, light and dark cell samples for protein extraction were taken from the cultures near the end of the last light or dark hour scheduled in the fixed L:D 12:12 cycle. The concentrations of all protein extracts were assessed with BCA Protein Assay Kit (KeyGen Biotech, Nanjing, China) and standardized by dilution. Aliquots of 60 μg protein extract were loaded onto 12% SDS-PAGE and transferred to polyvinylidene difluoride (PVDF) membranes (Merck Millipore, Darmstadt, Germany). Western blots for the presence or absence of GFP alone, expected fusion proteins and histone H3 (nuclear standard) in the samples of total, cytoplasmic, and nuclear proteins from each L:D cycle were probed by 1000-fold dilutions of anti-GFP Tag mouse monoclonal antibody (Invent Biotechnologies; Catalog No.: MAN2007) and anti-histone H3 rabbit polyclonal antibody (Abcam, Shanghai, China; Catalog No.: ab1791), respectively. The bound antibody was reacted with 5000-fold dilution of horseradish peroxidase (HRP) conjugated anti-rabbit antibodies (Boster, Wuhan, China; Catalog No.: BA1054) and visualized in a chemiluminescence detection system (Amersham Biosciences, Shanghai, China). Each blotting experiment was repeated three times, and a representative gel was presented.

### 2.3. Time-Course Western Blot Experiments

To reveal expression dynamics of Frq1, Frq2, and FRH in nucleus and cytoplasm, transgenic strains expressing *frq1*::*gfp* in Δ*frq2* and *frq2*::*gfp* in Δ*frq1* were constructed as aforementioned totransform the plasmid of each fusion gene into WT. As deleted from WT, *frh* was deleted from the TS1 expressing *frq1*::*gfp* and the TS2 expressing *frq2*::*gfp*, yielding the strains TS1::Δ*frh* and TS2::Δ*frh*. The submerged cultures of all transgenic strains in the backgrounds of WT (TS1–3), Δ*frq1*, Δ*frq2*, and Δ*frh* were initiated with 50 mL aliquots of a 10^6^ conidia mL^−1^ suspension in SDBY. After a 60-h light (L:D 24:0) or dark (L:D 0:24) incubation on a shaking bed (150 rpm) at 25 °C, the hyphal cultures of each strain were transferred to reversed dark or light condition for 24 h of shaking incubation. During the dark or light exposure, three cultures (replicates) were taken at 3-h interval for extraction of nuclear and cytoplasmic proteins with the aforementioned kits (Invent Biotechnologies). Western blots for time-course expression levels of Frq1::GFP, Frq2::GFP, and FRH::GFP in the nuclear and cytoplasmic protein extracts (60 μg loaded per lane) of the corresponding strains were probed with the anti-GFP antibody as aforementioned. Histone H3 probed by anti-H3 antibody and β-tubulin probed by anti-β-tubulin antibody (Invent Biotechnologies; Catalog No.: MAN1003) were used as nuclear and cytoplasmic standards respectively. The intensities of all western blots were quantified with ImageJ software (https://imagej.nih.gov/ij/) to assess time-course expression levels of Frq1::GFP and Frq2::GFP relative to histone H3 and β-tubulin in nucleus and cytoplasm respectively. The expression levels of Frq1::GFP in both WT and Δ*frq2* and those of Frq2::GFP in both WT and Δ*frq1* at the same time point of light or dark exposure were averaged to show nuclear dynamics and cytoplasmic stability of either Frq1 or Frq2 in a circadian day. The expression levels of Frq1::GFP and Frq2::GFP in all of the mentioned strains at the time points of both light and dark exposures were pooled to show a stability of total FRQ (mean of Frq1 and Frq2) level in nucleus relative to that in cytoplasm in a circadian day.

### 2.4. Assessment of Conidiation Capacity in Response to Daylight Length

The Δ*frq1*, Δ*frq2*, and Δ*frh* mutants and their control (WT and complemented) strains were grown on SDAY plates (9 cm diameter) spread with 100 μL aliquots of a 10^7^ conidia mL^−1^ suspension for culture initiation and incubated for 10 days at 25 °C in the L:D cycles of 0:24, 6:18, 12:12, 16:18, and 24:0, respectively. From the end of day 3 onwards, three plugs (5 mm diameter) were bored daily from each plate culture with a cork borer. The conidia on each culture plug were released into 1 mL of 0.02% Tween 80 via supersonic vibration. The concentration of conidia in the resultant suspension was assessed with a haemocytometer and converted to the number to conidia per unit area of plate culture (no. conidia cm^−2^). Aside from estimation of conidial yield, conidiation status in the 7-day-old SDAY cultures of *frq* mutants and control strains grown at 25 °C and L:D 12:12 was examined with scanning electronic microscopy (SEM). Trends of measured conidial yields (*y* in a scale of 10^7^ conidia cm^−2^) over daylight length (*l*_d_; h) and incubation time (*t*; h) were fitted to the modified logistic equation *y* = K/[1 + exp(a + b_1_*l*_d_ + b_2_*t*), where parameter K denotes conidiation capacity and parameters b_1_ and b_2_ represent hourly increase rates of conidial production depending on the two independent variables respectively.

### 2.5. Transcriptional Profiling

Aliquots (100 μL) of a 10^7^ conidia mL^−1^ suspension were evenly spread onto cellophane-overlaid SDAY plates for culture initiation. The cultures used for transcriptional profiling of *frq1* and *frq2* in the WT strain were incubated at 25 °C for 48–168 h at fixed L:D 12:12 or 72 h in the L:D cycles of 24:0, 18:6, 12:12, 6:18, and 0:24. The cultures used for transcriptional profiling of three CDP genes, the key UDA pathway gene *fluG*, and four photorecepor genes in the *frq* mutants and WT and of *frq1*, *frq2*, and the aforementioned genes in the *frh* mutants and WT were incubated for 72 h at 25 °C in different L:D cycles. Total RNAs were extracted from the WT cultures incubated 48–168 h in the fix L:D cycle or from the 72-h-old cultures of each strain in different L:D cycles using the RNAiso™ Plus Reagent (TaKaRa, Dalian, China), and reversely transcribed into cDNAs at 37 °C using the PrimeScript^®^ RT reagent kit (TaKaRa). For each strain, three cDNA samples (standardized by dilution) of each trea™ent (culture age or L:D cycle) were used as templates to assess transcript levels of target genes relative to γ-actin gene (internal standard) via real-time quantitative PCR (qPCR) with paired primers (Appendix A) under the action of SYBR^®^ Premix ExTaq™ (TaKaRa). A threshold-cycle (2^−ΔΔCt^) method was used to compute relative transcript levels of *frq1* and *frq2* in the WT cultures incubated 72–168 h with respect to the standard level in the 48-h-old culture at L:D 12:12 or 72 h in different L:D cycles with respect to the standard level at L:D 12:12, and of the mentioned genes in the 72-h-old cultures of the *frq* or *frh* mutants with respect to the WT standard in each L:D cycle.

## 3. Results

### 3.1. Sequence Features of Frq1, Frq2, and FRH in *B. bassiana*

In *B. bassiana*, Frq1 (964 aa) features an extremely large FRQ domain (pfam09241, residues 13–953) like both l-FRQ (989 aa) and s-FRQ (890 aa) in *N. crassa*, but structurally distinct from Frq2 (583 aa), which has two FRQ domains (residues 27–248 and 256–557) (Appendix A). Domain analysis at https://www.ncbi.nlm.nih.gov/Structure/ also reveals the presence of an LA motif RNA-binding domain (LAM, cd07323) in the central region of the FRQ domain of Frq1, l-FRQ, or s-FRQ but its absence in Frq2. Revealed by further domain analysis at http://smart.embl-heidelberg.de, the *N*-terminal region of Frq1, l-FRQ, or s-FRQ contains a coiled-coil domain that is essential for FRQ to form functional homodimers [31,32] but absent in Frq2. Moreover, an NLS motif (8 to 23 aa) and a small *C*-terminal region (17 aa) reported as an FRQ-FRH interacting domain (FFD) to form the FFC required for FRQ-WCC interaction [41] were found in the sequences of Frq1, Frq2, and the FRQ queries, but not revealed through the domain analyses. In Frq2, notably, the largest NLS serves as a bridge between the two FRQ domains. In phylogeny, Frq2 is also distinct from Frq1 and FRQ homologs found in other filamentous fungi, including insect, nematode and plant pathogens (Appendix A). Although a unique FRQ appears in most of the surveyed fungal genomes, two seemingly distinct FRQs also exist in *Hirsutella minnesotensis* (908 and 1628 aa) and *Cordyceps militaris* (962 and 908 aa). Surprisingly, *Fusarium oxysporum* V64-1 has five single FRQ domain-containing proteins (126 to 991 aa) but only two of them show relatively high sequence identity to the l- and s-FRQ. However, none of these fungal FRQ homologs contains two FRQ domains like Frq2 in *B. bassiana*. The differences of Frq1 and Frq2 in molecular size and structure implicate functional peculiarity of both in *B. bassiana*. The FRH orthologs of *B. bassiana* and *N. crassa* are similar to each other in molecular size and structure, share 70% sequence identity, but have a distinct C-terminal domain (Appendix A). Importantly, *frh* is nonessential for *B. bassiana* viability, providing a chance to clarify whether FRH plays an essential role in the feedback loop.

### 3.2. Expression of Frq1 and Frq2 Is Mutually Independent

Both *frq1* and *frq2* were similarly transcribed in the WT strain during a 168-h incubation on SDAY at the optimal regime of 25 °C and L:D 12:12 and reached a transcript peak at the end of 120-h incubation (Figure 1A). The two *frq* genes also showed similar transcript levels in the 72-h-old WT cultures grown at 25 °C in different L:D cycles with respect to the standard at L:D 12:12 (Figure 1B). Their transcripts decreased simultaneously with shortening daylight lengths. Shown in LSCM images, the GFP-tagged Frq1 (Figure 1C) and Frq2 (Figure 1D) fusion proteins were well expressed in the membrane dye-stained hyphae of two transgenic strains (TS1 and TS2) expressing *frq1*::*gfp* and *frq2*::*gfp* fusion genes in the WT strain at L:D 12:12. To clarify subcellular localization of either fusion protein, total, cytoplasmic and nuclear protein extracts isolated from the 3-day-old hyphal cultures of TS1 and TS2 grown at the optimal regime were probed with the anti-GFP antibody. As a result, expressed Frq1::GFP (131.8 kD; Figure 1E) and Frq2::GFP (92.7 kD; Figure 1F) were detected in both cytoplasmic and nuclear protein extracts irrespective of the cultures taken near the end of light or dark hour at L:D 12:12. Further, hyphae were collected from the 3-day-old cultures of TS1 and TS2 grown at 25 °C under continuous light (L:D 24:0) or dark (L:D 0:24) conditions, stained with the nuclear dye and analyzed with LSCM. Meanwhile, the same antibody was used to probe each fusion protein in total, cytoplasmic, and nuclear protein extracts isolated from the dark or illuminated cultures. The LSCM images presented a nuclear localization of Frq1::GFP only at L:D 24:0 and of Frq2::GFP only at L:D 0:24 (Figure 1G). Such a nuclear localization was also confirmed by the western blots of Frq1::GFP (Figure 1H) and Frq2::GFP (Figure 1I).

All of these data demonstrate an independent expression of Frq1 and Frq2. Therefore, the two FRQs are unlikely the isoform products of a single gene in *B. bassiana* like the *N. crassa* l-FRQ and s-FRQ formed by splicing the same gene at different positions [26,27,28]. Due to the opposite light and dark cues required for nuclear localization of the two fusion proteins, we speculate that nuclear accumulation levels of Frq1 and Frq2 in *B. bassiana* may fluctuate with reverse light and dark cues.

### 3.3. Frq1 and Frq2 Show Light-Dependent Opposite Dynamics in Nucleus and Light-Independent High-Level Stability in Cytoplasm

The above speculation was clarified by monitoring expression dynamics of Frq1::GFP and Frq2::GFP in nucleus and cytoplasm, respectively, through time-course western blot experiments with nuclear protein extracts, which were isolated from the hyphal cultures of TS1 and TS2 exposed 3–24 h to light after 60-h incubation at L:D 0:24 or to dark after 60-h incubation at L:D 24:0. The western experiments were performed due to a requirement of nuclear localization for FRQ to function in the *Neurospora* clock [24]. As a result, both Frq1:GFP and Frq2::GFP accumulated in cytoplasm at steadily high levels similar to each other irrespective of the time-course exposure to light or dark (Figure 2A). Relative accumulation levels of each fusion protein versus β-tubulin showed no significant variability (*P* ≥ 0.37 in *F* tests) over the time of either time-course exposure (Figure 2B). By contrast, nuclear Frq1::GFP accumulation was undetectable unless the TS1 cultures were exposed at least 6-h to light after continuous dark incubation, followed by a steady increase over the time of light exposure, and peaked in the last sample (Figure 2C). Reversely, nuclear Frq1::GFP accumulation peaked at the end of continuous light incubation and declined over the time of dark exposure to an undetectable level in the last two samples. Interestingly, Frq2::GFP in the extracts from the TS2 cultures showed nuclear dynamics opposite to those of Frq1::GFP during the period of time-course exposure to either light or dark cue. Consequently, nuclear accumulation levels of the two fusion proteins versus histone H3 fluctuated towards opposite directions during the time-course exposure to light or dark (Figure 2D). Their opposite dynamics in nuclei were reinforced by two or three nuclear protein samples containing undetectable Frq1::GFP and Frq2::GFP at the inverted time points of time-course exposure to light or dark. Further time-course western blots of Frq1::GFP expressed in Δ*frq2* and of Frq2::GFP expressed in Δ*frq1* also revealed steadily high levels of either fusion protein in cytoplasm and their opposite dynamics in nucleus (Figure 2E–H).

These results indicate that the opposite dynamics of Frq1 and Frq2 in nucleus are light dependent while their similar levels and high stability in cytoplasm are independent of light. The fact that two or three samples contained no nuclear Frq1::GFP or Frq2::GFP signal implicates at least 3- or 6-h disappearance of Frq1 or Frq2 in nucleus. The opposite nuclear dynamics represent phase-complementary rhythms and also implicate a steadily high level of total FRQ (pooled Frq1 and Frq2) in nucleus under light or in the dark.

### 3.4. Frq1 and Frq2 Are Essential for Rapid Conidiation in Different Photoperiods

The WT strain usually forms small zigzag rachises (conidiophores) for initiation of conidiation at the optimal regime as early as less than 72 h after culture initiation with conidia spread on SDAY, followed by formation of spore balls comprising plenty of conidia and rapid increase of such spore balls in size, density and quantity over the time of incubation (Appendix A). To explore why Frq1 and Frq2 coexist in *B. bassiana*, we monitored conidiation levels of two Δ*frq* mutants and their control strains during a 10-day incubation at optimal 25 °C in five different photoperiods (L:D 0:24, 6:18, 12:12, 16:18, and 24:0). Each strain showed a steady increase of conidial yield with both daylight length (*l*_d_) and incubation time (*t*). Conidial yields quantified from the cultures of the control strains at L:D 0:24 and 24:0 increased from ~0.3 × 10^7^ to ~1.5 × 10^7^ conidia cm^−2^ at 72 h and ~36 × 10^7^ to ~55 × 10^7^ conidia cm^−2^ at 168 h, and were stabilized at ~55 × 10^7^ conidia cm^−2^ from 192 h onwards irrespective of the photoperiod change (Figure 3A). Compared to the control strains, the Δ*frq1* and Δ*frq2* mutants suffered a great delay in conidiation and a sharp reduction in conidial yield; their conidiation defects were markedly worsened with shortening daylight length. Conidial yield in the presence of Frq2 in Δ*frq1* was not measurable before an incubation of 192 h at L:D 0:24 or 144 h at L:D 16:8 or 24:0, and increased from 4.5 × 10^6^ conidia cm^−2^ at L:D 0:24 to 87.6 × 10^6^ conidia cm^−2^ at L:D 24:0 by the end of a 10-day incubation. In the presence of Frq1 in Δ*frq2*, similarly, conidiation did not start before an incubation of 216 h at L:D 0:24 or 144 h at L:D 24:0, resulting in a yield increase from 2.1 × 10^6^ conidia cm^−2^ at L:D 0:24 to 61.6 × 10^6^ conidia cm^−2^ at L:D 24:0 on the last day. Shown in SEM images of conidiation status, the control strains produced plenty of conidia in a 7-day-old culture at L:D 12:12 (Figure 3B). By contrast, the formation of conidiating structures was just initiated in the same culture of Δ*frq1* but not in Δ*frq2*, the hyphae of which had not been differentiated yet. By the end of 10-day incubation at L:D 0:24, 6:18, 12:12, 16:18, and 24:0, conidial yield decreased by 99.6%, 99.4%, 98.7%, 92.6%, and 88.6% in Δ*frq1*, and 99.2%, 98.2%, 97.5%, 89.4%, and 83.9% in Δ*frq2* versus the WT strain, respectively. Modeling analysis of the yield trends over *l*_d_ and *t* revealed a significant dependence of conidial yield on both variables (see fitted curves and equations in Figure 3A). Particularly, the fitted conidiation capacity (parameter K) decreased by 69.6% in the presence of Frq2 in Δ*frq1* and 84.4% in the presence of Frq1 in Δ*frq2* with respect to the WT standard. However, biomass levels of both Δ*frq* mutants measured from the 7-day-old cultures in different photoperiods were similar to those of their control strains (Figure 3C), indicating no link of their severe conidiation defects to hyphal growth. Moreover, transcript levels of three CDP genes required for *B. bassiana* conidiation [7,8] and upstream *fluG* critical for the *brlA* activation [59] were largely reduced in the 3-day-old cultures of two Δ*frq* mutants versus WT at 25 °C and more reduced at L:D 0:24 than at L:D 24:0 (Figure 3D). By contrast, four photoreceptor genes, particularly *wc-1* and *wc-2*, in the Δ*frq* mutants were up-regulated more significantly at L:D 24:0 than at L:D 0:24. We also tried to construct double deletion mutants of *frq1* and *frq2* but failed in many attempts, hinting at an essentiality of *frq1* or *frq2* for cell viability when the other is absent.

All of these results highlight that both Frq1 and Frq2 are required for continuity and rapidity of conidiation in different photoperiods and maximization of conidial yield for survival and dispersal of *B. bassiana* in host habitats. Notably, the *l*_d_-dependent conidiation was compromised in Δ*frq2* more than in Δ*frq1*, implicating higher sensitivity of Frq2 than of Frq1 to a photoperiod change likely due to a shorter period of its presence in the nucleus. The repressed expression of the three CDP genes, particularly *brlA*, suggests essential roles of Frq1 and Frq2 in the CDP activation required for the fungal conidiation.

### 3.5. FRH Is Required for Nuclear Dynamics and Cytoplasmic Stability of Frq1 and Frq2

To explore how Frq1 and Frq2 are controlled to activate the CDP genes for support of rapid non-rhythmic conidiation in different photoperiods, we examined changes of nuclear and cytoplasmic Frq1::GFP and Frq2::GFP in the presence or absence of *frh*. First, FRH::GFP expressed in TS3 (Figure 4A) accumulated exclusively in nucleus at a highly stable level under light or in the dark (Figure 4B) and also in time-course response to dark after 60-h light incubation or to light after 60-h dark incubation (Figure 4C). Also shown in Figure 4C, deletion of *frh* in TS1 or TS2 resulted in undetectable or residual accumulation level of either Frq1::GFP or Frq2::GFP in both nucleus and cytoplasm during the time-course exposures. Consequently, both opposite nuclear dynamics and cytoplasmic stability of Frq1 and Frq2 were abolished in the absence of *frh*. Second, deletion of *frh* had no effect not only on conidiation capacity at 25 °C in different photoperiods but also on conidiation speed in daylight lengths of ≤12 h (Figure 4D). Third, transcripts of both *frq1* and *frq2* were also reduced to residual levels in the 3-day-old Δ*frh* cultures with respect to the WT standard either under light or in the dark (Figure 4E). In contrast with up-regulated expression in the absence of *frq1* or *frq2*, four photoreceptor genes were differentially repressed in Δ*frh*. Particularly, transcript levels of *wc-1* and *wc-2* encoding essential components of the WCC were reduced more than those of *vvd* and *phy*. By contrast, transcript levels of the three CDP genes and upstream *fluG* were not significantly affected in the Δ*frh* mutant showing normal conidiation capacity. Both phenotype and transcript changes observed in Δ*frh* were well-restored by targeted *frh* complementation.

All of these data indicate an indispensability of FRH for transcription and translation of *frq1* and *frq2* and also its involvement in transcriptional activation of *wc-1* and *wc-2*. In Δ*frh*, abolished Frq1 and Frq2 accumulation in both nucleus and cytoplasm concurred with normal expression of the CDP genes and little change in conidiation capacity under light or dark conditions. These strongly suggest some other factors (likely WCC components) involved in the Frq1-FRH and Frq2-FRH interactions to mediate non-rhythmic conidiation in *B. bassiana* and also implicate that the CDP activation is independent of Frq1 and Frq2 when both of them are out of the FRH control.

## 4. Discussion

As presented above, FRH is required for stable accumulation of Frq1 and Frq2 in cytoplasm and opposite dynamics of both FRQs in nucleus. Such opposite nuclear dynamics in all time-course exposure experiments can be used to outline opposite nuclear dynamics of Frq1 and Frq2 in a circadian day of different photoperiods (Figure 5A) and a total FRQ (pooled Frq1/2) level that is relatively stable in nucleus (Figure 5B) and high enough to persistently take part in nuclear events, such as transcriptional activation of the CDP genes to support continuous conidiation of *B. bassiana* in daytime and nighttime. Taken these results into account, we propose a model for two FRQs to co-function in activating the CDP genes in the presence of FRH in *B. bassiana*, as illustrated in Figure 5C. Beginning from nighttime, the most activated Frq1 works alone for ~9 h in the Frq2-absent nucleus, co-works with Frq2 for another 9 h, and then disappears from the nucleus for ~6 h. Starting from daytime, the most activated Frq2 works alone for ~6 h in the Frq1-absent nucleus. During the co-working period, the relative importance of each FRQ correlates with its accumulation level in the nucleus. Either Frq1 or Frq2 works alone during the period of its high nuclear level, requires the other’s cooperation when its nuclear accumulation decreases to an insufficient level, and terminates its nuclear role when the other accumulates to a sufficient level in the nucleus for initiation of independent work. The roles of Frq1 and Frq2 in the nucleus are related at least to activation of the CDP genes through certain checkpoint(s) unknown at present, as unveiled by transcriptional repression of the CDP genes leading to severe conidiation defects in Δ*frq1* and Δ*frq2*. The CDP activation is unlikely related to the roles of Frq1 and Frq2 in the cytoplasm, in which total FRQ level was steadily high in Δ*frq1* or Δ*frq2* and abolished only in Δ*frh*. Due to a nucleus-specific localization, we hypothesize that FRH can interact with Frq1 or Frq2 only in the nucleus to form possible Frq1-FRH or Frq2-FRH complex in *B. bassiana* as does the unique FFC to further interact with WCC in *N. crassa* [35,37,38,39,40]. This hypothesis is supported by the presence of FFD in both Frq1 and Frq2 and also by the fact that both opposite nuclear dynamics and cytoplasmic stability of Frq1 and Frq2 were abolished in the absence of *frh*. These findings uncover a novel, but more complicated, circadian system that works in *B. bassiana* and support the fungal asexual life. Particularly, transcriptional repression of the CDP genes in Δ*frq1* and Δ*frq2* and their normal expression in Δ*frh* demonstrate a novel link of the CDP activation to the circadian system, in which WCC acts as a transcription factor and interacts with FFC [20,21,22]. We speculate that Frq1 and Frq2 interacting with the unique FRH dispensable for cell viability might serve as phase-complementary oscillators of the same clock in *B. bassiana*. The CDP genes might be regulated through an interaction of the Frq1-FRH or Frq2-FRH complex or both complexes with the WCC. These warrant further studies.

A complexity of the circadian system learned from *Neurospora* is characterized by the feedback loop consisting of the activating positive and repressing negative arms, as reviewed in [18,19]. While the roles of two co-working FRQs and unique FRH have been clarified in this study, it remains unknown how they individually or collectively interact with the critical WCC to operate the whole system in *B. bassiana*. Particularly, two core component-coding genes of the WCC to activate *frq* expression in *N. crassa* [20,21,22] were up-regulated in the absence of *frq1* or *frq2* but repressed in the absence of *frh*, which led to not only repressed transcription of both *frq1* and *frq2* but also abolished accumulation of either FRQ in both nucleus and cytoplasm under light or in the dark. These results highlight an essential role for FRH in the feedback loop and suggest much more complicated protein-protein interactions in the circadian system of *B. bassiana* than those learned from the *Neurospora* model.

Nuclear FRQ is degraded through progressive hyper-phosphorylation critical for the timekeeping of the *Neurospora* clock, as reviewed in [60]. It is the ATPase activity of FRH that attenuates the hyper-phosphorylation of FRQ mediated by casein kinase 1a and is related to functional inactivation of FRQ before its degradation [45,61]. Although it remains unclear how FRH interacts with other enzymes for timed FRQ hyper-phosphorylation and degradation, FRH was confirmed in this study to stabilize in nucleus the opposite dynamics of Frq1 and Frq2 to a total FRQ level high enough to activate the CDP genes for non-rhythmic conidiation at every moment of each circadian day and sustain the asexual cycle of *B. bassiana* all the year round. Like *B. bassiana*, many filamentous fungi undergo asexual cycle alone and lack phenotypes to report circadian rhythms. Our findings have opened a door to in-depth dissection of the clock components to interact with Frq1, Frq2, and/or FRH in *B. bassiana*, and also shed light upon the discovery of novel circadian systems in other filamentous fungi by monitoring nuclear expression dynamics of their FRQ homologs.

## Figures and Tables

**Figure 1 cells-09-00626-f001:**
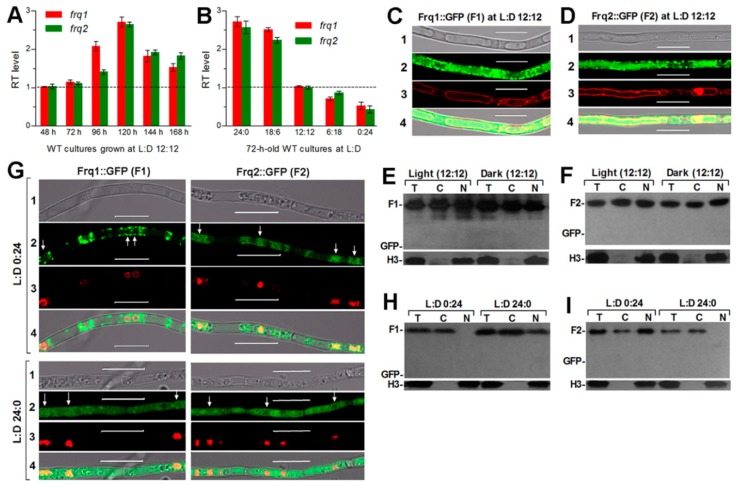
Transcriptional expression and subcellular localization of Frq1 and Frq2 in *B. bassiana*. (**A**,**B**) Relative transcript (RT) levels of *frq1* and *frq2* over the time (h) of cultivation at the optimal regime of 25 °C and L:D 12:12 with respect to the standard at 48 h and in the 72-h-old cultures grown in different L:D cycles with respect to the standard at L:D 12:12, respectively. All WT cultures were initiated by spreading 100 μL aliquots of a 10^7^ conidia mL^−1^ suspension on SDAY plates. Error bars: SD from three cDNA samples analyzed via qPCR with paired primers (Appendix A). (**C**,**D**) Laser scanning confocal microscopic (LSCM) images (scale = 10 μm) for subcellular localization of Frq1::GFP (F1) and Frq2::GFP (F2) in transgenic hyphae, which were collected from the 72-h-old cultures of a 10^6^ conidia mL^−1^ suspension in SDBY shaken at 25 °C and L:D 12:12 and stained with the membrane-specific dye FM4-64. (**E**,**F**) Western blots for F1 (131.8 kD) and F2 (92.7 kD) detected by anti-GFP antibody in total (T), cytoplasmic (C) and nuclear (N) protein extracts, which were isolated from the 72-h-old SDBY cultures of transgenic strains taken near the end of light or dark hour at L:D 12:12. (**G**) LSCM images (scale = 10 μm) for subcellular localization of F1 and F2 in the hyphae collected from the 72-h-old SDBY cultures at L:D 0:24 and L:D 24:0, respectively, and stained with the nuclear dye DAPI. (**H**,**I**) Western blots for F1 and F2 in the protein extracts isolated from the SDBY cultures grown at L:D 0:24 and L:D 24:0, respectively. Rows 1−4 in C, D, and G denote bright, expressed, stained, and merged views of the same field. Arrows in C indicate locations of nuclei. Each lane in E, F, H, and I was loaded with 60 μg protein extract. Nuclear standard: histone H3 probed by anti-histone H3 antibody. Note that GFP alone (27 kD) is undetectable with the anti-GFP antibody, indicating a fusion of Frq1 or Frq2 to GFP as expected.

**Figure 2 cells-09-00626-f002:**
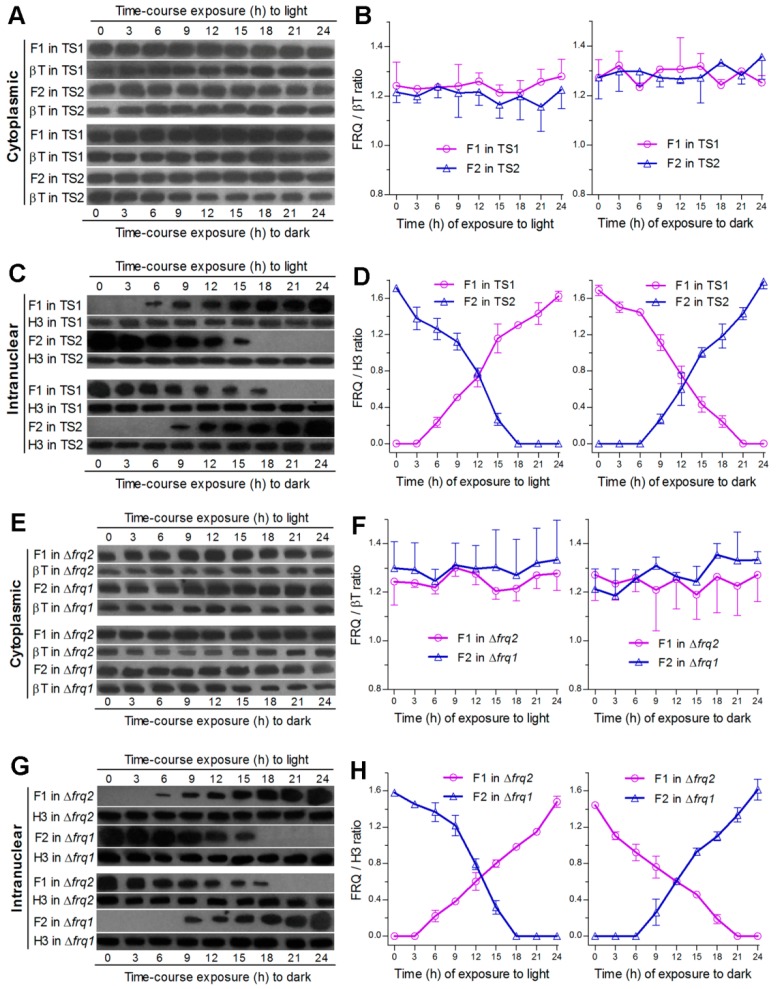
Intracellular expression dynamics of Frq1 and Frq2 in time-course responses to light and dark cues. (**A**–**D**) Time-course western blots and relative expression levels of Frq1::GFP (F1) and Frq2::GFP (F2) in the cytoplasm and nucleus of the WT-based transgenic strains TS1 and TS2 respectively. (**E**–**H**) Time-course western blots and relative expression levels of F1 in the cytoplasm and nucleus of Δ*frq2* and of F2 in the cytoplasm and nucleus of Δ*frq1*. All cytoplasmic and nuclear protein extracts were isolated from the hyphal cultures exposed 0–24 h (3-h interval) to light after continuous 60-h dark incubation (upper panels) or to dark after continuous 60-h light incubation (lower panels) of a 10^6^ conidia mL^−1^ suspension in SDBY at 25 °C. Each fusion protein was probed by anti-GFP antibody. Nuclear standard: histone H3 probed by anti-H3 antibody. Cytoplasmic standard: β-tubulin (βT) probed by anti-β-tubulin antibody. Each lane was loaded with 60 μg protein extract. Each presented gel is a representative of three western replicates. Relative expression level of each fusion protein in cytoplasm or nucleus was estimated as the ratio of its blot intensity over that of β-tubulin or H3. Error bar: standard deviation (SD) of the mean from three replicates.

**Figure 3 cells-09-00626-f003:**
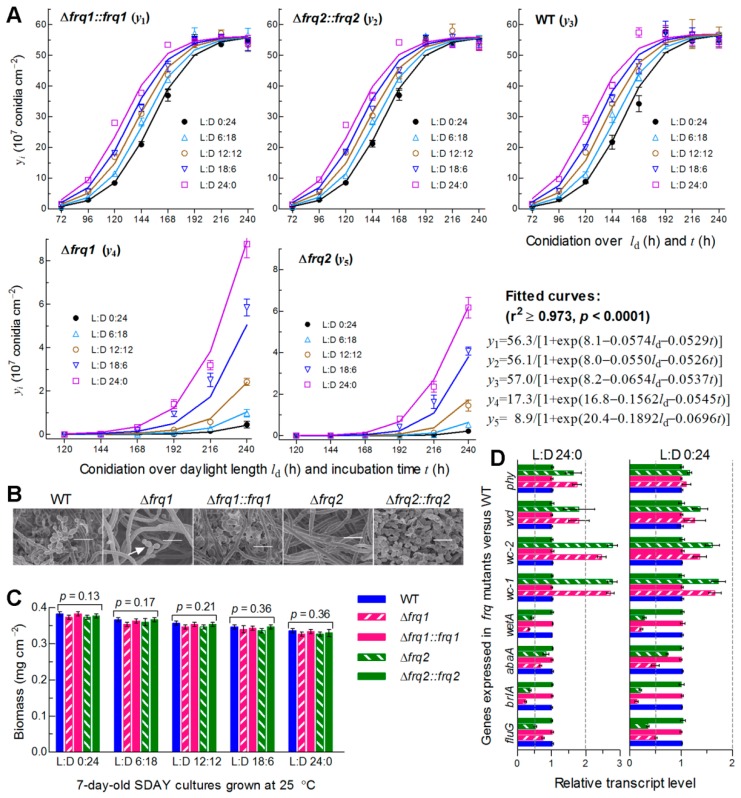
Essential roles of Frq1 and Frq2 in non-rhythmic conidiation of *B. bassiana*. (**A**) Conidial yields (symbols) measured from the SDAY cultures of *frq* mutants and WT during a 10-day incubation at 25 °C in five different L:D cycles and those fitted to daylight length (*l*_d_) and incubation time (*t*). The fitted curves and equations indicate a significant dependence of conidial yield (*y* in a scale of 10^7^ conidia cm^−2^) on *l*_d_ and *t* (*P* < 0.0001) and a great difference in conidiation capacity between Δ*frq* mutants and control strains. (**B**) Scanning electronic microscopic (SEM) images (scale = 5 μm) for conidiation status of indicated strains in the 7-day-old cultures grown at 25 °C and L:D 12:12. Note that early conidiating structures were present in Δ*frq1* (arrowed) but absent in Δ*frq2*. (**C**) Biomass levels measured from the 7-day-old SDAY cultures grown at 25 °C in five photoperiods. (**D**) Relative transcript levels of related genes in the 3-day-old SDAY cultures of *frq* mutants versus WT grown at 25 °C under light (L:D 24:) and in darkness (L:D 0:24). The dashed line denotes a significance for one-fold repression or up-regulation. All SDAY cultures were initiated by spreading 100 μL of a 10^7^ conidia mL^−1^ suspension per plate (9 cm diameter). Error bar: SD of the mean from three replicates (**A**,**C**) or cDNA samples (**D**) analyzed through qPCR with paired primers (Appendix A).

**Figure 4 cells-09-00626-f004:**
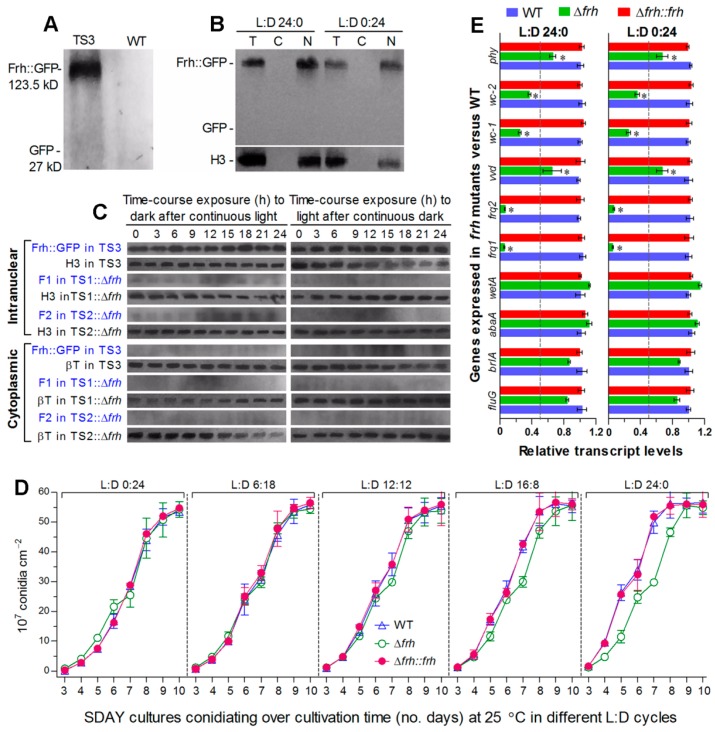
Nuclear FRH is required for Frq1 and Frq2 to function in nucleus. (**A**,**B**) Western blots for accumulation levelsof GFP and FRH::GFP in the total protein extracts of transgenic strain (TS3) and WT cultures at L:D 12:12 and of FRH::GFP in the total (T), cytoplasmic (C), and nuclear (N) protein extracts of 3-day-old TS3 cultures grown in SDBY at 25 °C and L:D 24:0 or 0:24. (**C**) Time-course western blots for accumulation levels of FRH::GFP in TS3 and of Frq1::GFP (F1) and Frq2::GFP (F2) in TS1::Δ*frh* and TS2::Δ*frh* strains. Nuclear and cytoplasmic protein extracts were isolated from the hyphal cultures exposed 0–24 h (3-h interval) to dark after 60-h light incubation (left panels) or to light after 60-h dark incubation (right panels) of a 10^6^ conidiamL^−1^ suspension in SDBY at 25 °C. Each lane was loaded with 60 μg protein extract and probed by anti-GFP antibody. Nuclear standard: histone H3 probed by anti-H3 antibody. Cytoplasmic standard: β-tubulin (βT) probed by anti-β-tubulin antibody. Note that accumulation level of FRH::GFP in TS3 was steadily high in nucleus during the time-course exposure but undetectable in cytoplasm and that deletion of *frh* led to residual or undetectable levels of F1 and F2 in both nucleus and cytoplasm. (**D**) Conidial yields measured from the SDAY cultures of *frh* mutants and WT during a 10-day incubation at 25 °C in five L:D cycles. (**E**) Relative transcript levels of *frq1*, *frq2*, four photoreceptor genes and four developmental activator genes in the 3-day-old SDAY cultures of *frh* mutants versus WT grown at 25 °C under light and dark conditions. The dashed line denotes a significance for one-fold repression. Error bar: SD of the mean from three replicates (**D**) or cDNA samples (**E**) analyzed through qPCR with primers (Appendix A).

**Figure 5 cells-09-00626-f005:**
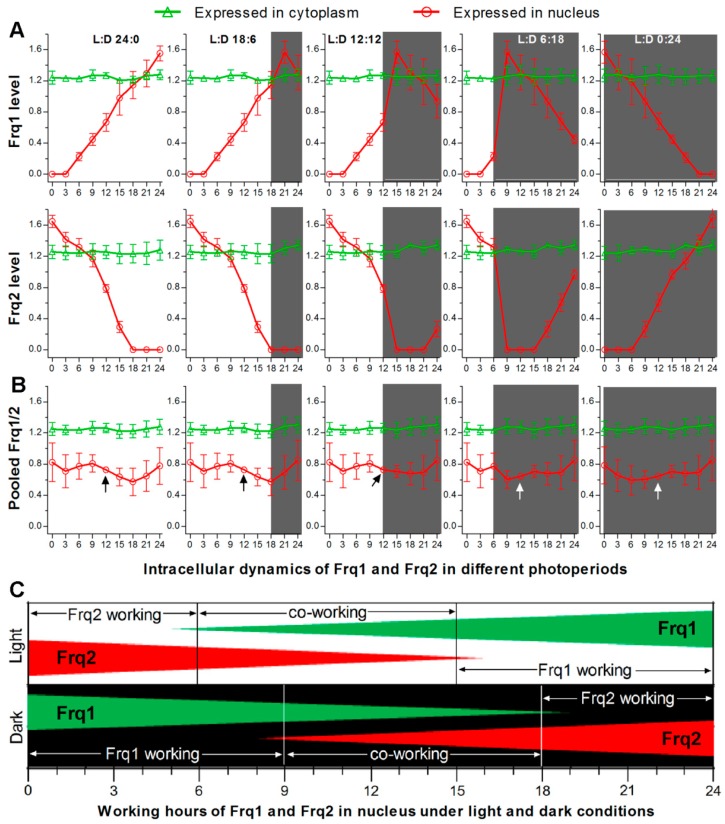
Proposed model for Frq1 and Frq2 to co-function in *B. bassiana*. (**A**) Possible changes in nuclear expression levels (rhythms) of Frq1 and Frq2 in a circadian day of different photoperiods. Nuclear and cytoplasmic accumulation levels of Frq1::GFP or Frq2::GFP in TS1 or TS2 and those of each fusion protein in the absence of the other *frq* (shown in Figure 2) were averaged on each sampling occasion during time-course exposure to light or dark. Error bar: SD of the mean from six replicates. (**B**) Dynamics of total FRQ (mean of Frq1 and Frq2) levels in nucleus and cytoplasm. Note that accumulation level of total FRQ in nucleus fluctuates in a narrow range and is most stable at the median time point (arrowed) of a circadian day irrespective of photoperiod change. Error bar: standard error (nuclear) or SD (cytoplamic) of the mean from 12 replicates. (**C**) Proposed model for Frq1 and Frq2 to function on alternate duty in the nucleus in the presence of FRH. Either Frq1 or Frq2 works alone for a period, co-works with the other for another period, and then disappears in nucleus irrespective of a day beginning from daytime or nighttime.

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
