# Peer review of "Opposite Nuclear Dynamics of Two FRH-Dominated Frequency Proteins Orchestrate Non-Rhythmic Conidiation in Beauveria bassiana"

_cells, 2020, doi:10.3390/cells9030626_

Round 1

Reviewer 1 Report

Title: Opposite Nuclear Dynamics of Two FRH-Dominated Frequency Proteins Orchestrate Non-Rhythmic Conidiation in Beauveria bassiana

Authors: Sen-Miao Tong, Ding-Yi Wang, Qing Cai, Sheng-Hua Ying and Ming-Guang Feng.

Recommendation: Major Review

                In this work, the authors characterize the role of the two FRQ and the FRH orthologs of the insect pathogen Beauveria bassiana. After a preliminary characterization of the sequences of the two FRQ orthologs and their evolution, the authors observe an opposite nuclear accumulation dynamics in light or dark, with FRQ1 being accumulated in nuclei in light (but absent in the dark), and FRQ2 being accumulated in nuclei in the dark (but absent in light). In a deeper analysis, it is described how FRQ1 increases nuclear accumulation with time in light, and FRQ2 shows the same behavior with time but in the dark. The same dynamics is described for each FRQ protein in the null mutant of the other.

                The absence of any frq gene decrease the quantity of conidia produced by B. bassiana, and despite the fact that both FRQ1 and FRQ2 require FRH activity for their appropriate nuclear dynamics, it seems that conidia production and the expression of key genes controlling asexual development is not affected in the Δfrh strain. Thus, the authors do not elucidate how, being dependent on FRH, FRQ1/2 control conidiation.

                There is a huge amount of work in this manuscript and, in my opinion, is of interest for the readers of Cells. However, there is a main point not shown by the authors. They show that the nuclear dynamics of FRQ proteins is dependent on FRH activity, but based on other model organisms they assume that the control of FRQ dynamics by FRH is enabled by their interaction (see the model in Figure 5, line 500). A pull-down assay would confirm this hypothesis. Second, the nuclear localization of FRH::GFP is not shown in the manuscript, and would be interesting to confirm the nuclear accumulation described by fractionation experiments. Finally, there are several minor errors that should be addressed before acceptance, and the quality of the language should be improved. I attach a pdf copy of the manuscript with my comments and corrections. My recommendation is Major Review.

Author Response

Please see attached our responses.

Reviewer 2 Report

The submitted paper “Opposite nuclear dynamics of two FRH-dominated frequency proteins orchestrate non-rhythmic conidiation in Beauveria bassiana” from Tong et al. addresses different interesting and important questions on the role of these proteins in an enthomopathogenic fungus.

The experiments shown are well conducted and robust.

My main comment is about the organization of the figures, which seems to me too loaded with a lot of different information. This is mainly the case for Figures 1 and 3 and to a lesser extent for Figure 4. It would seem clearer for Figure 1 to separate the LSCM observations (C, D and G) and explaining lines 1 to 4. For figure 4, graphs C and D could also form another figure and would enlarge the barely visible image B.

As minor comment, Latin names must be italicized.

Author Response

Please see attached our responses.

Round 2

Reviewer 1 Report

The authors have replied to all my queries.